# The Effect of Age, Gender, and Job on Skin Conductance Response among Smartphone Users Who are Prohibited from Using Their Smartphone

**DOI:** 10.3390/ijerph17072313

**Published:** 2020-03-30

**Authors:** Hsiu-Fen Hsieh, Hsin-Tien Hsu, Pei-Chao Lin, Yu-Jen Yang, Yu-Tung Huang, Chih-Hung Ko, Hsiu-Hung Wang

**Affiliations:** 1School of Nursing, College of Nursing, Kaohsiung Medical University, Kaohsiung 807, Taiwan; hsiufen96@gmail.com (H.-F.H.); hthsu@kmu.edu.tw (H.-T.H.); pclin@kmu.edu.tw (P.-C.L.); 2Department of Medical Research, Kaohsiung Medical University Hospital, Kaohsiung Medical University, Kaohsiung 807, Taiwan; 3Department of Nursing, Tzu Chi University of Science and Technology, Hualien 97071, Taiwan; starthas@gmail.com; 4 Center for Big Data Analytics and Statistics, Chang Gung Memorial Hospital, Taoyuan City 333, Taiwan; anton.huang@gmail.com; 5Department of Department of Psychiatry, Kaohsiung Medical University Hospital, Kaohsiung Medical University, Kaohsiung 807, Taiwan; chihhungko@gmail.com; 6College of Medicine, Kaohsiung Medical University, Kaohsiung 807, Taiwan; 7Taiwan Nurses Association, Taipei 10681, Taiwan

**Keywords:** problematic smartphone use, skin conductance response, anxiety, gender, withdrawal

## Abstract

The smartphone is a widely used and rapidly growing phenomenon worldwide, and problematic smartphone use is common in our society. This study’s objective was to examine the gender difference of baseline and post-intervention skin conductance response (SCR) among smartphone users and explore the relationships among problematic smartphone use level, anxiety level, and SCR changes by evaluating SCR, the Zung Self-Rating Anxiety Scale score, and the Chinese version of the Smartphone Addiction Inventory (SPAI) score in a one-group baseline and post-test design. Sixty participants were recruited from two communities, and data were collected from April to June 2017. There was a significant difference in terms of SCR changes between young males and old males and between young females and old females. Additionally, the SCR changes in young females were significantly greater than those in young males with twofold mean difference. This study provides strong evidence supporting the effectiveness of SCR measurement for assessing problematic smartphone use (PSU) anxiety when users are in a withdrawal-like state. The SCR measurement can help healthcare providers identify cases with risk factors of PSU for early intervention.

## 1. Introduction

Global smartphone use continues to increase, with the number of smartphone users already being at 4.15 billion worldwide in 2015 and expected to rise to 4.78 billion by 2020 [1]. The proportions of smartphone users are especially high in some countries such as South Korea (88%), Australia (77%), the United States (72%) [1], and Taiwan (82.7%) [2]. One of the most useful smartphone functions is that people can be connected to the internet as they wish, enabling the sending or receiving of e-mail, browsing of social networks, information searches, or enjoyment of online games anytime, anywhere [3]. According to a report released by the Taiwan Network Information Center (TWNIC) [4], the current percentage of the Taiwanese population that surfs the internet with smartphones is approximately 87.4%. Furthermore, a survey conducted by the Institute for Information Industry [5] indicated that about 81% of Taiwanese people cannot leave the house without their smartphones and that their dependence on smartphones is the highest in the world. In addition, TWNIC [4] conducted a global mobile internet survey and found that most Taiwanese people used mobile phones to access the internet every day, with an average daily use time of 204 minutes. 

Internet addiction is widespread and has been studied extensively in recent years [6]; many internet-related conditions or disorders, such as internet gaming disorder, have been included in the International Classification of Diseases (ICD-11) and the Diagnostic and Statistical Manual of Mental Disorders (DSM-5) [7]. Like internet addiction, problematic smartphone use (PSU) could be categorized as a behavioral addiction with excessive use of specific apps, such gaming apps or social media apps. Smartphones provide great conveniences in communication and entertainment and many other functions. Research suggests that PSU is associated with physical, psychological, and social functioning [8,9]. Withdrawal is the process of cutting out or decreasing the use of addictive substances, such as drugs or alcohol, or behaviors, such as gambling [10]. As a condition included in the DSM-5 requiring further study, Internet Gaming Disorder (IGD) refers to persistent and recurrent internet gaming associated with clinical impairment or distress [7]. Criterion 2 of IGD refers to “withdrawal symptoms following the removal of Internet games.”

Use of social media (e.g., Facebook, Instagram, etc.) is currently one of the most popular leisure activities owing to its multiple functions, such as interacting with friends, meeting others based on shared chatting, sharing or creating pictures/videos, playing games, etc. [11]. There are numerous social media apps. LINE is the most commonly used social media app in Asian countries. About 86.5% of the smartphone users in Taiwan currently use the LINE app, and this proportion is much higher compared to that of other social media apps, such as Facebook, Twitter, etc. [9]. 

### 1.1. Model of Problematic Mobile Phone Use

The model of problematic mobile phone use was developed by Bianchi et al. in 2005 [12]. Problematic mobile phone use is a concept of behavioral addiction that includes the core components of addictive behaviors, such as tolerance, withdrawal, emotional instability, etc. The problematic mobile phone use model consists of many pathways of addiction, such as the excessive reassurance pathway which includes “established risk factors,” “privileged application (app),” “type of problematic use,” and “symptoms or behaviors” [13]. We modified the components of the excessive reassurance pathway for use in our study as a pathway of problematic smartphone use [13]. 

### 1.2. Established Risk Factors

Some smartphone users experience problematic smartphone use (PSU) through the excessive and compulsive use of their smartphones. These people are usually characterized by an inability to control their smartphone use, and by their social or occupational impairment [14,15]. Females use smartphones mostly for social connections, while male users are more likely to use them for game programs and work-related purposes [16]. It was shown that adolescents would be at higher risk of smartphone addiction as compared with adults because they are yet to develop self-control in using a smartphone [17]. However, the total amount of time spent using smartphones, as well as the dependency on these devices, is higher for men than women. In particular, due to the increase in everyday smartphone use, the degree of reliance on the smartphone is increasing [18]. In addition, women exhibit a greater addiction to smartphones, especially on weekends, which affects their sleep patterns [19]. Another study found that the risk factors for college students’ PSU included female gender, using the internet, drinking, and anxiety [20].

### 1.3. Symptoms or Behaviors

Increased heart rate and sweat gland activity were found to be the key physiological indicators of craving for an addictive substance [21]. PSU, like substance addiction, is regarded as the inability to control the use of a smartphone, and physiological features of craving can be seen in people with PSU when they are in withdrawal status. In severe cases, smartphone use might lead to significant financial, physical, and psychological consequences, as well as social functional impairment [15]. Pain, discomfort, or functional impairment of associated organs among problematic smartphone users have been well documented in many studies [22,23]. In addition, some studies have shown that people with PSU experience poorer quality of sleep and higher levels of depression and anxiety than those without PSU [16,23]. 

A previous study has shown that skin conductance response (SCR) changes measured immediately following emotional fluctuation [24] and demonstrated that SCR could be considered an index to measure emotional change [24,25]. A previous study observed higher SCR reactivity in anxious people than those without anxiety [26]. Anxiety is one of the main symptoms of withdrawal, and it has been reported that severe problem internet users exhibited increased levels of anxiety with correlated changes in their SCR after internet use was stopped [27]. 

Owing to a lack of research on the relationship between change in SCR and anxiety among people with PSU use when they are in withdrawal status, we planned to use SCR as a measurement to examine the difference in problematic social media use-related withdrawal anxiety among smartphone users with potential PSU. Clinical workers can early refer these individuals to an adequate unit for early intervention to prevent addiction. The study objectives are, therefore, to (a) examine the gender and age difference in baseline SCR measurements and post-SCR among smartphone users and (b) explore the relationships among level of PSU, changes in SCR (withdrawal anxiety of problematic social media use), and level of anxiety (SAS). 

The hypotheses are: (a) those with problematic social media use have higher post-test SCR; (b) younger participants and females tend to have higher levels of PSU, anxiety, and mean difference of SCR than the older and male participants; (c) those with anxiety (SAS scores > 50) or PSU (SPAI scores > 58) have higher mean difference of SCR. 

## 2. Materials and Methods

### 2.1. Participants

Our study sample was drawn from two communities in Kaohsiung City in Southern Taiwan. Recruitment fliers were posted around the two participating communities. A total of 71 participants were eligible for this study. Eleven participants did not complete all questionnaires or skin conductance measures. Inclusion criteria for the participants were those who (a) were older than 20 years of age; (b) used a smartphone with internet and social media software; and (c) were motivated and available to participate in this study. The exclusion criteria for the participants were those who (a) had alcoholism or drug dependence; (b) had any pre-existing psychiatric disorders, such as anxiety or depression; (c) had a history of brain injury; and (d) had been diagnosed with palmar hyperhidrosis or had higher raw SCR signals (based on the SCR testing).

### 2.2. Ethical Considerations and Data Collection

This research was approved by the Institutional Review Board (IRB No. KMUHIRB-E (I)-20170016) of a medical center. Data were collected from April 2017 to June 2017. The principal investigator (PI) explained the purpose and processes of this study to all of the potential participants and asked for their willingness to participate. Written informed consent was obtained from all participants, who were also informed that their participation was voluntary, any personal information would remain confidential, and they had the right to withdraw their participation at any time without providing a reason. 

### 2.3. Physiological Measurements

The SCR was collected by a biofeedback machine with the ProComp InfinitiTM system (Thought Technology Ltd., Montreal, Quebec, Canada), which was installed on a laptop [28]. The SCR is an index of sympathetic nervous system activation and emotional arousal [29]. To measure SCR, a small electrical potential is applied between two electrodes strapped or taped to two fingers, and the amount of current conducted between the electrodes is measured. The electrode strap must be fastened tight enough around a finger so that the electrode surface is in contact with the finger pad, but not so tight that it limits blood circulation. Researchers cleaned the electrodes with an alcohol wipe between participants to ensure accurate measurements. The SCR results were analyzed using the ProComp Infiniti SC analysis module (Thought Technology Ltd.). Fast Fourier transformation was used to transform the skin conductance level data to provide an SCR by means of a pass digital filter (0.05 Hz–1 kHz) [28]. The amplifier was connected to the computer through an optical connection, and the peak-to-base index was used as an indicator of SCR (e.g., the difference between the maximum value recorded in the time frame of 8 s post stimulus and the baseline calculated on a trial-by-trial basis) [30,31].

### 2.4. Self-Report Data

All participants completed a questionnaire that collected demographic characteristics, PSU levels, and anxiety levels. 

Anxiety state. The Zung Self-Rating Anxiety Scale (SAS) was developed by Zung [32] to measure levels of anxiety. The scale has been employed in a number of cross-cultural studies and translated into other languages with nearly 20 versions. The SAS test is a self-administered questionnaire consisting of 20 items, each rated on a scale from 1 to 4 points (1 = a little of the time to 4 = most of the time). Total scores range from 20 to 80, and higher scores indicate higher levels of anxiety. The raw score was multiplied by 1.25 to yield the final total score ranging from 25 to 100. A total score below 50 was considered normal, 50 to 59 indicated mild anxiety, 60 to 69 indicated moderate anxiety, and scores above 70 indicated severe anxiety. The reliability of the SAS was demonstrated by Cronbach’s α with a value of 0.72, and construct validity was 0.89.

PSU. The Chinese version of the Smartphone Addiction Inventory (SPAI) was designed and developed on the basis of the Chen Internet Addiction Scale (CIAS) [33]. It contained a four-factor structure: compulsive behavior, functional impairment, withdrawal, and tolerance [34], and we used this questionnaire to examine our participants’ PSU. Total SPAI scores range from 26 to 104, with higher scores indicating higher levels of smartphone addition. The diagnostic cut-off of 64/65 was most suitable to discriminate cases of smartphone addiction from diagnosis-negative cases, with maximum diagnostic accuracy (74.6%). The screening cut-off point of 57/58 presented an acceptable sensitivity (71.4%) with maximum diagnostic accuracy (70.5%), and Youden index (0.417); according to the study of Ko and colleagues (2005), it also indicated an optimal cut-off point for screening possible cases of smartphone addiction [33]. The Cronbach’s α for the overall scale was 0.94, and for the four factors (“compulsive behavior (SPAI_Com),” “functional impairment (SPAI_FI),” “withdrawal (SPAI_Wit),” and “tolerance (SPAI_Tol)” it was 0.87, 0.88, 0.81, and 0.72, respectively. A two-week test–retest reliability (intra-class correlations) of the overall scale and 4 subscales ranged from 80 to 91 [35]. 

### 2.5. Research Design and Procedure

This was a single-group with baseline and post-test research design. SCR and structured questionnaires were both measured. The SCR protocol included self-guided muscle relaxation, diaphragmatic breathing, paced breathing, and pursed-lips breathing which were used by Lin et al. [28]. The LINE app (LINE Corp., Tokyo, Japan) is an instant communication freeware that provides a variety of services, such as a digital wallet (LINE Pay) and video on demand (LINE TV). Moreover, LINE provides real-time voice and video calls, and users can exchange texts, photos, video, audio, and stickers to express emotions. Our study employed this LINE app to produce social stimuli. 

At the beginning of our study, the PI created a LINE link with every participant, and we used this app as the only tool to create social stimuli, make phone calls, and send the messages. All tests were performed while the participants were in a sound-attenuated and temperature-controlled room. Initially, both sound and vibration notifications were turned on before social media stimulation, and each smartphone was placed in a box on a table near the sofa. The sound and vibration of the smartphone could easily be perceived by each participant.

After a 10 min rest, an SCR sensor was placed on the participant, and the physiological signals were calibrated for about 2 min before baseline measurement. Thereafter, the raw SCR signals were measured and recorded for a period of 5 min, and then transformed into the baseline SCR indices. The post-test SCR measurements were obtained during a 5 min window, and participants were prohibited from using their smartphone during this period. This period was divided into five 1 min sub-periods, and each participant received three messages and one call averaging 30 sec, followed by a 30 sec break before the next set of stimuli. Such stimuli were presented in every minute during this 5 min period with a total of 15 messages and 5 calls from the PI. During this 5 min period, the participants might also receive messages or calls from their friends or families. This design was intended to create a withdrawal-like state with accumulation of anxiety. While the smartphone sounds and vibrations were easily detectable, the participants were prohibited from reading or answering the messages and calls during this period. SCR indices were compared between baseline and post intervention. After the intervention, the PI gave back the smartphone to every participant, and they were allowed to use their smartphone and rest for a period of time. Each participant completed the SAS and SPAI questionnaires following the break (Figure 1).

### 2.6. Statistical Analyses

The sample size was calculated using G*Power statistical software version 3.1(Heinrich-Heine-Universität, Düsseldorf, Germany) [36] with an effect size of 0.35, significance level of 0.05 for a two-tailed test, and power of 0.8. A sample size of 59 was calculated to be adequate. The data analyses were performed using IBM SPSS version 19.0 (IBM Corp., Armonk, NY, USA). We compared the demographic characteristics between the male and female participants. We compared the SCR (baseline SCR, post-test SCR, and mean difference of SCR), subscales of PSU level (SPAI_Com, SPAI_Wit, SPAI_Tol, and SPAI_FI), and anxiety level between gender (male and female) and age (young and old) of our participants, and we also compared the SCR between the non-anxiety and anxiety groups and between the non-PSU and PSU groups. Bivariate analysis was used to identify the association between anxiety and PSU, and between age and job. Changes in SCR were transformed into statistical data and figures for further analysis using BioGraph Infiniti software (version 6.0.4; Thought Technology Ltd., Montreal, Quebec, Canada) [28]. The effects of SCR were also examined by comparing the SCR indices pre and post intervention. Statistical analyses were performed using SPSS 19.0. A *p* value < 0.05 was interpreted as statistically significant.

## 3. Results

### 3.1. Demographic Data

A total of 60 participants met the inclusion criteria and completed the research procedures: 31 females (51.7%) and 29 males (48.3%) with a mean age of 26.87 (standard deviation = 8.44) years and their age ranged from 21 to 36 years (male: 21–36 years; female: 21–34 years). We divided all participants into a 2 × 2 design by age (young/old) and gender (M/F). The young males and young females checked their smartphone more frequently than the old males and old females, and young females had a significantly higher frequency of checking their smartphone than old females. In addition, there was a significant difference in jobs and marital status as all of the young males and young females were college students and they were single/never married (Table 1). 

### 3.2. Comparison of the Skin Conductance Response between Anxiety and PSU Groups

The PSU score was positively correlated with level of anxiety (*p* = 0.033). In addition, we divided our participants into non-PSU and PSU groups, and non-anxiety and anxiety groups, and student and employee groups. In our study, 26.7% of participants had anxiety. By SPAI cut-off points 57/58 and 64/65, we found that 68.3% of our participants had PSU and 30% (n = 18) had smartphone addiction. We analyzed their relationship based on baseline SCR, post-test SCR, and mean difference of SCR by t-test. We found that there was no significant difference between non-anxiety and anxiety groups, and between non-PSU and PSU groups in terms of SCR, whereas the student group had significantly higher levels of SCR than the employee group (post-test SCR and mean difference SCR) (*p* < 0.001) (Table 2).

### 3.3. The Skin Conductance Response, PSU, and Anxiety among Young Males, Old Males, Young Females, and Old Females

In this study, there was no significant gender and age difference at baseline SCR (*p* > 0.05). We divided our participants into the young (<24 years old) and old (≥24 years old) groups, according Kelley’s rule of 27% and 73% [37]. Then we divided all participants into a 2 × 2 design for analyzing M/Y, M/O, F/Y, and F/O (M = males; F = Females; Y = Young; O = Old), and we found that there was a significant difference in terms of SCR changes between young males and old males (*p* = 0.020) and between young females and old females (*p* = 0.002). In addition, the mean difference of SCR in young females was significantly greater (mean = 24.40) than that in young males (mean = 11.11) with a twofold difference, but there was no significant association between PSU and anxiety (Table 3). We further analyzed the relationship among age, gender, and mean difference of SCR, and it showed that old males had a significantly lower mean difference of SCR than old females (*p* < 0.05) (Figure 2). Spearman’s correlation was conducted for exploring the association between age and job, and it showed that age was strongly positively correlated with job (*p* < 0.001). 

### 3.4. The Relationship among Gender, Job, and mean Difference of SCR

After controlling for the effect of gender, there was no significant interaction effect between job and gender on mean difference of SCR (F (1,56) = 0.744, *p* = 0.392), but both of the main effects of gender (*p* = 0.037) and job (*p* = 0.017) on mean difference of SCR were statistically significant (Table 4), and male employees had significantly lower mean difference of SCR than female employees (*p* < 0.05) (Figure 3).

## 4. Discussion

Under the model of problematic mobile phone use (PSU), we identified the established risk factors (female, age, change of SCR (PSU withdrawal-like), students), symptoms, or behaviors (problematic social media use, anxiety, withdrawal). Approximately 68.3% of our participants exhibited higher rates of PSU, ranging from 10% to 35%, than those in previous studies [13,38]. A study conducted in Saudi Arabia reported that 48% of undergraduate university students had PSU [39], and we found that most of our participants had PSU. 

One possible reason is that increasingly more smartphone gaming apps are available for download especially by male users. It has been reported that 62% of smartphone owners install a game within one week after receiving the phone, and smartphone games currently account for more than 43% of total time spent on smartphones [40]. In addition, young females spend longer talking to their friends by smartphone or are drawn to social media more readily, which may be a direct result of how people are socialized [41]. 

### 4.1. Established Risk Factors 

In our study, we found that young age, females, and being a college student are risk factors for developing PSU. Our study results indicated a significant change in SCR after LINE stimulation when participants were requested not to read or answer smartphone calls or text messages, and young males and young females showed a significantly greater increase in SCR than old males and old females. In addition, the change of SCR in young females was significantly greater than that in young males with twofold mean difference. This result was similar to previous studies that showed a greater increase in SCR among internet abusers when they stopped using the internet [26], and adolescents are a high-risk group for smartphone addiction [17,42]. 

Furthermore, young participants, especially young female participants, spent more time using their smartphones than the old participants in our study, and there was a strong positive association between age and job. This result suggested that being a college student itself is a risk factor developing PSU. This result also corresponded to a previous study [43] suggesting that female students spend significantly more time on their phones per day than male students.

This result can be explained by some studies, where female students were found to spend more time talking to their friends by their smartphone and used social media for interpersonal relationships [23,41], and male students spent more time on playing games [44]. In other words, students spent time on using smartphone for a variety of purposes, such as social communication or playing games, and this might be the reason why students become problematic smartphone users. In summary, young age, female, and being a college student are risk factors for developing PSU with higher mean difference of SCR. 

### 4.2. Symptoms or Behaviors

In our study, the average SAS scores were below the cut-off point (>50) of anxiety. In addition, there was no significant difference in anxiety between males and females and between the young and old groups. This result was not compatible with previous studies [19,26], suggesting that the young or female tend to have PSU with anxiety rather than the old or male. Only about one-fourth of our participants had anxiety, and the possible reason is that all participants had a period of time for at least five minutes to relax and read the texting messages or missed calls, and their intense anxiety which was measured by SCR during the experimental period might have relieved before filling in the questionnaires so that they had relatively lower SAS scores. This also explained why some of our participants were not aware of their PSU. 

Smartphones have become an essential part of our lives today. Users could begin exhibiting withdrawal symptoms after only a short period of separation from their smartphones, and withdrawal symptoms following a sudden cessation of using smartphone was a key feature of PSU. People may experience distress if deprived of their smartphone for some time, regardless of the reason. The possibility of missed text messages or calls while the smartphone is switched off also makes users anxious, irritable, and insomnolent [45,46]. Our result was relevant to previous studies suggesting that students had a higher risk to have PSU, and they spent more time on using their smartphone [23,41]. This could explain that our student participants, especially female students, likely had withdrawal anxiety with a significantly higher post-test SCR when they were in a withdrawal-like circumstance. 

In sum, our participants had an increase in post-test SCR when we created a withdrawal-like circumstance, especially, students had a significantly greater increase in post-test SCR and mean difference of SCR than employees. In our study, objective data (change in SCR) was more sensitive than subjective data (self-report questionnaires) for assessing anxiety, and SCR measurement can be completed in a short period of time to detect anxiety and withdrawal reaction. We suggest that other researchers can use this objective tool to obtain more accurate data for evaluating anxiety-related problems in future studies.

There were many limitations of this study. First, the research design used only a single group; there was no control group. The sample size was small, and the design did not ensure a balanced sample of subjects in each group (student/employee). In the future, a longitudinal study with a larger sample size, balanced subjects, and a control group could be designed and conducted for a deeper understanding of PSU and associated factors. Second, we had exclusion criteria, but did not use the validated tools to screen for substance abuse or major mental disorders. Third, withdrawal symptoms depend on the type of substance/behavior and duration of use; physical and psychological characteristics; and the withdrawal process used. However, we did not measure these factors.

## 5. Conclusions

The novelty of our study is that we used a smartphone social media app as a stimulus maker and we measured participants’ withdrawal anxiety by means of SCR under a withdrawal-like circumstance. Regarding the change in SCR after social media stimulation, a significant difference was observed by age, job, and gender when they were requested not to read or answer smartphone calls or text messages, and young females showed a significantly greater increase in SCR than young males. Our study indicated that worrying over missed text messages or phone calls has a significant impact on people with PSU. 

In addition, PSU was positively correlated with anxiety, which implies that PSU is a risk factor of mental health problems. In clinical settings, healthcare providers can use SCR measurements for early detection of cases with PSU and design intervention programs for those people. Furthermore, more attention should be paid to the screening and treatment of concurrent mental problems for people with PSU.

## Figures and Tables

**Figure 1 ijerph-17-02313-f001:**
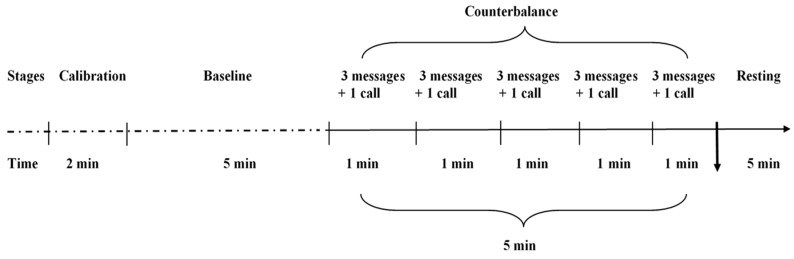
Experimental protocol.

**Figure 2 ijerph-17-02313-f002:**
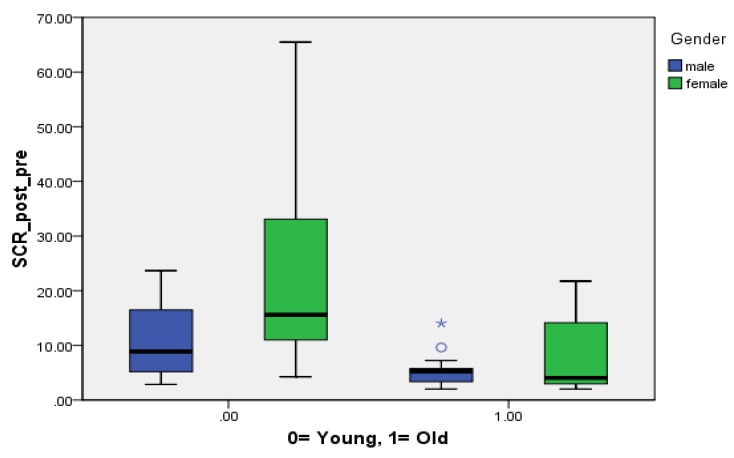
Age and gender on mean difference of SCR. (★) *p* < 0.05; (o) outlier.

**Figure 3 ijerph-17-02313-f003:**
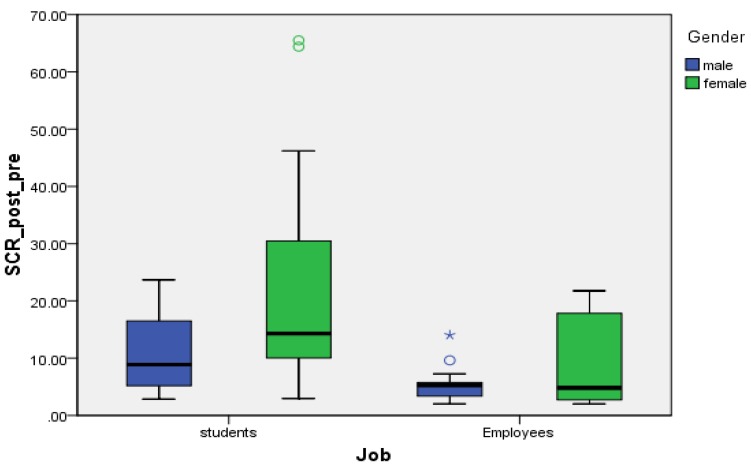
Job and gender on mean difference of SCR. (★) *p* < 0.05; (o) outlier.

**Table 1 ijerph-17-02313-t001:** Comparison between gender on the demographics and smartphone use/behavior.

	M/Y n = 13 (48.3)		M/O n = 16 (48.3)			F/Y n = 21 (51.7)		F/O n = 10 (51.7)		
Variables	M/n	SD(%)	M/n	SD(%)	*p*	M/n	SD(%)	M/n	SD(%)	*p*
Job					<0.001 ***					<0.001 ***
No (students)	13	(100)	0	0 %		21	(100)	3	(30.0)	
Yes (employees)	0	(0)	16	(100)		0	(0)	7	(70.0)	
Marital status					<0.001 ***					<0.001 ***
Single, never married	13	(100)	7	(43.8)		21	(100)	5	(50.0)	
Married	0	(0)	9	(56.3)		0	(0)	5	(50.0)	
How often smartphone was checked					0.582					0.021 *
Every 10 min	2	(15.4)	1	(6.3)		3	(14.3)	0	(0)	
Every 30 min	8	(46.1)	7	(43.7)		12	(42.9)	5	(50)	
Every 1 h	4	(30.8)	5	(31.3)		7	(33.3)	0	(0)	
Every 2 h	1	(7.7)	4	(25.0)		2	(9.5)	5	(50)	
Duration of smartphone use per day					0.351					0.092
Less than 1 h	2	(15.4)	0	(0)		0	(0.0)	0	(0.0)	
1–3 h	2	(15.4)	5	(31.3)		4	(19.0)	4	(40.0)	
3–5 h	7	(53.8)	8	(50.0)		13	(61.9)	2	(20.0)	
>5 h	2	(15.4)	3	(0)		4	(19.0)	4	(40.0)	

Note. M = Male; F = Female; Y = Young (<24 years old); O = Old (≥24 years old); M = mean; SD = standard deviation; * *p* < 0.05, *** *p* < 0.001.

**Table 2 ijerph-17-02313-t002:** The skin conductance response between non-anxiety and anxiety groups, between non-PSU and PSU groups, and between student and employee groups.

	Non-Anxiety (N = 44) 73.3%		Anxiety (N = 16) 26.7%			Non-PSU (N = 19) 31.7%		PSU (N = 41) 68.3%			Student (N = 37) 61.7%		Employee (N = 23) 38.3%		
Variables	Mean	SD	Mean	SD	*p*	Mean	SD	Mean	SD	*p*	Mean	SD	Mean	SD	*p*
B-SCR	0.34	0.37	0.45	0.40	0.318	0.39	0.36	0.36	0.39	0.778	0.41	0.45	0.31	0.20	0.231
Post-SCR	14.01	12.98	14.46	13.16	0.916	13.90	14.07	14.24	12.53	0.933	18.51	16.36	7.09	5.85	<0.001 ***
SCR_d	0.55	0.79	0.42	0.68	0.562	0.39	0.69	0.58	0.78	0.390	18.10	16.05	6.78	5.74	<0.001 ***

Note. SD = standard deviation; SCR = skin conductance response; B-SCR = baseline skin conductance response; Post-SCR = post-test skin conductance response; SCR_d = mean difference SCR; *** *p* < 0.001.

**Table 3 ijerph-17-02313-t003:** The skin conductance response, subscales of SPAI, and anxiety among young males, old males, young females, and old females.

	M/Y (N = 13)		M/O (N = 16)			F/Y (N = 21)		F/O (N = 10)		
Variables	Mean	SD	Mean	SD	*p*	Mean	SD	Mean	SD	*p*
Baseline SCR	0.35	0.24	0.30	0.17	0.547	0.50	0.55	0.26	0.24	0.203
Post-test SCR	11.46	7.53	5.68	3.15	0.020 *	24.90	18.49	8.51	8.05	0.002 *
SCR_d	11.11	7.44	5.38	3.03	0.020 *	24.40	18.10	8.25	7.90	0.002 *
SPAI										
SPAI_Com	18.38	3.59	17.44	3.44	0.476	19.86	4.61	19.20	3.12	0.687
SPAI_Wit	15.54	2.44	14.31	2.77	0.222	16.71	3.89	16.20	2.39	0.704
SPAI_Tol	7.23	1.79	7.25	1.88	0.978	7.86	2.22	7.60	0.84	0.646
SPAI_FI	16.62	4.23	19.06	4.71	0.157	18.95	4.95	19.40	3.57	0.801
Anxiety	46.23	8.69	45.31	8.86	0.782	42.54	12.51	42.88	8.66	0.931

Note. M = Male; F = Female; Y = Young (<24 years old); O = Old (≥24 years old); SD = standard deviation; SCR = skin conductance response; baseline SCR = skin conductance response before receiving text messages or calls; post-test SCR = skin conductance response after receiving text messages or calls; SCR_d = mean difference SCR; SPAI_Com = compulsive behavior; SPAI_FI = functional impairment; SPAI_Wit = withdrawal; SPAI_Tol = tolerance; * *p* < 0.05.

**Table 4 ijerph-17-02313-t004:** Analysis of Covariance (ANCOVA) results gender (males and females), and job (students and employees).

Source	Type III Sum of Squares	df	Mean Square	F	*p*
Corrected Model	2898.512a	3	966.17	6.068	0.001
Intercept	7213.356	1	7213.36	45.301	0.000
Gender	726.582	1	726.58	4.563	0.037
Job	962.393	1	962.39	6.044	0.017
Gender * Job	118.522	1	118.52	0.744	0.392
Error	8916.96	56	159.231		
Total	23,174.35				
Corrected Total	11,815.47

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
