# Peer review of "The Effect of Age, Gender, and Job on Skin Conductance Response among Smartphone Users Who are Prohibited from Using Their Smartphone"

_ijerph, 2020, doi:10.3390/ijerph17072313_

Round 1
Reviewer 1 Report
This manuscript describes the gender difference of skin conductance response (SCR) among smartphone users at baseline and post-test. It also explores the relationships among problematic smartphone use (PSU) level, anxiety level, and SCR changes.
I feel the paper contributes to an area of smart phone use research where more information is needed. The manuscript is well structured and written. Overall, the methods used are well chosen and clearly reported. They appear appropriate for the analyses carried out and the data and screening tools used also appear to be fit for purpose. The authors have also recognized the limitations of the study as well as discuss possible future directions.
I only have a few comments/suggestions to the authors:
-This is an interesting and timely topic, and I find the research design and objective quite exciting. However, I believe the authors could highlight more in the paper why this type of research is currently needed; why is it important to examine the gender differences?
-What was the age range of the participants? It would be interesting to see the age of the youngest vs the oldest participant.
-The authors mention the study employed the LINE app to produce social stimuli. It remained a bit unclear how this was executed. Were the phone calls and text-messages during testing sent using this particular app?
-What was the rationale for choosing “education”, “religious belief”, “marital status”, and “monthly living expenses” as independent variables?
Author Response
Reviewer 1: I only have a few comments/suggestions to the authors:
-This is an interesting and timely topic, and I find the research design and objective quite exciting. However, I believe the authors could highlight more in the paper why this type of research is currently needed; why is it important to examine the gender differences?
A:
|
We measured participants’ anxiety under a withdrawal-like status by SCR and anxiety questionnaire which was usually used to measure anxiety in most researches, so it is worth examining gender differences between SCR and anxiety questionnaire. The existence of gender difference was well documented in the issues of internet addiction, smartphone addiction, and other addictive behaviors, but no research on change of SCR of smartphone users between gender under a withdrawal-like status. |
-What was the age range of the participants? It would be interesting to see the age of the youngest vs the oldest participant.
A:
|
The age of our participants ranges from 21 to 36 years old (Table 1). Thank you for your recommendation and reminding, we divided our participants into the young (< 24 years old) and old (≥ 24 years old) groups, according Kelley’s rule of 27% and 73%, there was a significant difference on SCR (post-test SCR and mean difference SCR) between these two groups (lines 255-257 and Table 3). |
-The authors mention the study employed the LINE app to produce social stimuli. It remained a bit unclear how this was executed. Were the phone calls and text-messages during testing sent using this particular app?
A: Indeed, our study employed the LINE app to produce social stimuli. At the beginning of our study, the PI created a Line link with every participant, and we used this app as the only social stimuli maker to make phone calls and send the messages (Lines 184-186).
-What was the rationale for choosing “education”, “religious belief”, “marital status”, and “monthly living expenses” as independent variables?
A: Thank you for your reminding, we deleted these variables which are not closely related to our purpose of study after panel discussion (Table 1).
Reviewer 2:
This manuscript studies the use of skin conductance response (SCR) for measuring anxiety with problematic smartphone use (PSU). Data is analyzed to study relationships between pre/post SCR changes and level of PSU, anxiety, and gender. PSU is a growing problem with nearly ubiquitous adoption of smartphones worldwide. There is demonstrated evidence in the literature to support the motivation for this project, namely to assess the levels of PSU/PSMU by gender and anxiety levels. SCR has been widely used to measure stress and anxiety since the 1960’s so the technology is well-developed and well-understood. The techniques deployed in these experiments is straightforward and a classical pre/post-test intervention design is implemented to study changes in SCR before and after removal of smartphones.
Q1: It would seem the consecutive nature of the stimulus is not realistic with day-to-day operation (15 texts + 5 calls in 5 minutes). The analysis conducted is very descriptive in nature and in places incomplete.
A:
Our study need a biofeedback machine for SCR measurement, and every participant has his/her own job, it’s really not easy to carry out a day-to-day operation. In addition, based on the study of Lin et al., we aimed to create a withdrawal-like status during this 5-min period with counterbalance, and continuously measured participant’s SCR during this period.
Q2: Explanation is needed as to why the authors used a different SPAI cutoff level (57/58) for this study vs. clinically/diagnostically accepted levels (64/65).
A: Thank you for your recommendation. Since our purpose is to measure the SCR of and to early detect people with potential of problematic social media use, not for diagnosing those who had become smartphone addiction, we adopted 57/58 as SPAI cutoff point. The clinical workers can early refer those people with problematic social media use to an adequate unit for early intervention to prevent them from becoming addiction (Lines 111-114).
Q3:
As the manuscript focuses on SCR, there seems to be limited analysis of SCR values based upon each of the key parameters (anxiety, PSU, etc).
The interpretation and discussion of results may be improved with additional analysis. The key drawback of the manuscript is that the results seem to confirm what is already available in the literature without much innovation nor potential impact.
A: We have analyzed the relationship of SCR values, anxiety, and PSU (Lines 237-238 and Table 2).
Introduction: The introduction and review of the literature is sufficient. Some specific suggestions include:
1-1) This reviewer would suggest that acronyms be minimized wherever possible to enhance readability. The terms PSU, PMPU (pg 2, ln 68) and PSMU (pg 3, ln 103) are closely related and seem to be used interchangeably.
1-1) A:
Thank you for your recommendation. These terms cannot be used interchangeably for their different meaning, and to make it clearer, we will use the full name of PMPU and PSMU.
The model of problematic mobile phone use (PMPU) was developed prior to the mature development and wide use of smartphones, it’s a specific term and we will keep its full name.
Problematic social media use (PSMU) here is specifically for problematic “social media” use.
1-2)Discussion of the LINE social media app (pg 2, ln 58-66) seems out of place as it describes tools used for the experiments (i.e. methodology) more than literature review. Suggest moving to Materials and Methods section.
1-2)A: Thank you for your recommendation, we have moved this paragraph to Materials and Methods section (Lines 181-184).
1-3)As a point of distinction, is there a difference between phone use (i.e. text, phone calls) vs “social media”. The researchers broadly groups point-to-point interactions such as phone/texts with social media which broadly refers to larger social networks (i.e. Facebook, Twitter, etc).
1-3)A: There are numerous social media apps, LINE is the most commonly used social media app in Asian. About 86.5% of smartphone users in Taiwan currently use the LINE app, and this proportion is much higher than other social media apps, such as Facebook, Twitter, etc (Lines 62-67 ).
Materials and Methods:
2-1)As mentioned previously, description of the stimuli tool (LINE) should be moved to the Methods section of the paper, perhaps under “research design and procedure”.
2-1)A: Thank you for your recommendation, we have moved this paragraph to the part “research design and procedure” of the section of Methods (Lines 62-67).
2-2) Should an exclusion criterial include anxious subjects. It would seem those with high levels of anxiety may bias results.
2-2)A: Anxious subjects were excluded by our exclusion criteria: pre-existing psychiatric disorders, such as anxiety disorder, depressive disorder, etc (Lines 126-127).
2-3) Experimentally speaking, some explanation of the research design methodology is warranted. The design of 3 consecutive text messages with only ½ sec interval seems insufficient to allow subjects to return to baseline levels. What is the spacing between the time of the text messages and the phone call stimulus.
The repetition interval (1 min) for the 3 texts + 1 call seems insufficient to allow subjects to return to baseline. Without this, it seems that the research design seems to target accumulation of anxiety over time. Further, it seems that the design of 15 texts and 5 calls within a 5-minute period is not a realistic stimulus that is representative of normal smartphone use. It would seem there is a missed opportunity to measure SCR after each set of stimuli (3 text + 1 call).
2-3)A: Thank you for your comments.
First, the interval between two sets of stimuli is 1/2 min, not 1/2 sec, we used the wrong word (Lines 196-197).
Second, this 5-min period was divided into five 1-min subperiod which contained stimuli for 30 sec with 3 messages and 1 call, then followed by a break for 30 sec before the next subperiod. Such design is for counterbalance of stimuli, not for returning to baseline (Lines 194-197).
Third, we have measured participants’ baseline SCR before this period, and we kept giving stimuli to every participant during this 5-min period (Lines 192-193 and Figure 1).
Participants were prohibited from using their smartphone during this 5-min period, and this design is for creating a withdrawal-like status with accumulation of anxiety (Lines 198-200).
2-4) The composition of subject enrollment seems to be ad-hoc in nature and not matched. As such, there seems to be an older cohort of males than females with p values nearing significance (0.068). This reviewer wonders if unmatched age of subjects by gender influences the results. This concern is exacerbated by the proportion of females in the group who are students (77.4%) vs males. Perhaps the correlation to SCR is influenced by age (3rd variable) which is not studied here rather than by gender.
2-4)A: Thank you for your recommendation and reminding, we divided our participants into the young (< 24 years old) and old (≥ 24 years old) groups, according Kelley’s rule of 27% and 73%, there was a significant difference on SCR (post-test SCR and mean difference SCR) between these two groups. Your recommendation is really valuable and feasible, we found that age plays an important role in our study (lines 255-257 and Table 3).
2-5)On pg 4, ln 161, it states the “diagnostic cut-off of 64/65” is used to assess smartphone addiction. It is unclear then why a SPAI cut-off value of 57/58 is used for assessing PSU. This lower cut-off presumably has a lower sensitivity (71%) and accuracy (70.5%). This cut-off value is not well justified and seems rather arbitrary.
2-5)A: Thank you for your recommendation. The SPAI cut-off value of 57/58 was used for assessing PSU in our study, and it was documented that this cut-off point is an optimal cut-off point for screening possible cases of smartphone addiction (Chen, Weng, & Su, 2003). We used the SPAI cut-off value of 57/58 was for early detecting people with the potential of problematic use.
Results:
3-1)In general, t values presented is not particularly useful since this depends on the DOF and other factors. Perhaps only p-values are needed in tables.
3-1)A: Thank you for your recommendation. We have deleted t values.
3-2)Table 1 is hard to read and perhaps can be better organized into different categories. The table combines many different categories (demographics, income level; smartphone use, some that may be extraneous to the research question (religious belief? Income?). Perhaps the table should be divided into 2.
3-2)A: Thank you for your recommendation, we have organized Table 1 and divided it into two tables. In addition, we have deleted some variables which are not closely related to our study (Table 1).
3-3)Much of the information in Table 2 (SPAI) is repeated from Table 1. Suggestion to remove from Table 1. There seem to be much more variability in the SPAI results for males vs females. This should be explained. What accounts for the significantly larger std dev?
3-3) A: Thank you for your recommendation, we have added the variables of anxiety and SPAI in the Table 3. There were some cases with extreme SPAI values which resulted in larger std dev.
3-4)In Table 3, the “note” includes acronyms that are not used in the table and should be removed. It seems this note is more appropriate for Table 4.
3-4)A: Thank you for your recommendation, we have revised the Table 3 and Table 4.
3-5)The post-test SCR stdev for females is quite high compared to the pre-test values indicating a large range in outcomes due to intervention. Some discussion of this variance warrants additional analysis. Do some females exhibit increased SCR while others do not?
3-5) A: Thank you for your reminding, we checked our data again and found that some females exhibited obviously increased post-test SCR, and some did not increase much, and this led to a large SD [Table 3 and Figure 1].
3-6) As most of the study relates to SCR measures of anxiety and PSU, there seems to be a dearth of analysis based upon SCR and these other factors (anxiety, gender, etc). It would be helpful if Table 3 (or another table) also included SCR values for anxious vs. non-anxious and low/high SPAI subjects. Aggregation of all subjects regardless of PSU/anxiety levels makes it hard to assess key effects and is not particularly useful.
3-6) A: Thank you for your recommendation, we have added two tables containing Table 2 and Table 3 and analysis among the variables which you suggested
3-7)If clinically applicable SPAI thresholds are used to determine PSU, dues Table 4 change significantly? It would seem the selection of 57/58 (this study) vs. 64/65 (clinical) is rather arbitrary and not rigorous. “Note” for Table 4 copied twice.
3-7) A: Thank you for your recommendation.
The SPAI cut-off value of 64/65 was clinically used for assessing smartphone addiction. It’s an important cut-off value, but this cut-off value was not suitable for our study owing to different purpose. The SPAI cut-off value of 57/58 was used in our study to avoid missing those people with potential of problematic use.
3-8)There seem to be significant differences in employment level of the cohort and perhaps this may affect findings (in addition to or in place of gender). Perhaps an additional factor affecting PSU and SCR is age… and maturity. These analyses are not presented and is a missed opportunity of this study. It would seem the analysis is therefore incomplete.
3-8)A: Thank you for your recommendation, we have added some variables and examined the relationship among these variables. The analysis was added in the table and manuscript to make it more complete (Table 2, Table 3 and Figure 2).
4-1)Justification for utilization of lower SPAI scores to categorize PSU is warranted. This may influence results and findings.
4-1)A: Thank you for your recommendation.
The SPAI cut-off value of 64/65 was clinically used for assessing smartphone addiction. It’s an important cut-off value, but this cut-off value was not suitable for our study owing to different purpose. We used cut-off vale 57/58 was for people with problematic smartphone use (PSU), not only those with smartphone.
4-2)It seems the discussion and conclusions are drawn from incomplete data and perhaps influenced more by the research design than the actual phenomena. Additional analysis of the data may strengthen support for the key findings.
4-2)A: Additional analyses were completed, and the results were shown in Table 2, Table 3 and Figure 2. We also revised our discussion and conclusion.

Reviewer 2 Report
This manuscript studies the use of skin conductance response (SCR) for measuring anxiety with problematic smartphone use (PSU). Data is analyzed to study relationships between pre/post SCR changes and level of PSU, anxiety, and gender. PSU is a growing problem with nearly ubiquitous adoption of smartphones worldwide. There is demonstrated evidence in the literature to support the motivation for this project, namely to assess the levels of PSU/PSMU by gender and anxiety levels. SCR has been widely used to measure stress and anxiety since the 1960’s so the technology is well-developed and well-understood. The techniques deployed in these experiments is straightforward and a classical pre/post-test intervention design is implemented to study changes in SCR before and after removal of smartphones. It would seem the consecutive nature of the stimulus is not realistic with day-to-day operation (15 texts + 5 calls in 5 minutes). The analysis conducted is very descriptive in nature and in places incomplete. Explanation is needed as to why the authors used a different SPAI cutoff level (57/58) for this study vs. clinically/diagnostically accepted levels (64/65). As the manuscript focuses on SCR, there seems to be limited analysis of SCR values based upon each of the key parameters (anxiety, PSU, etc). The interpretation and discussion of results may be improved with additional analysis. The key drawback of the manuscript is that the results seem to confirm what is already available in the literature without much innovation nor potential impact.
Introduction: The introduction and review of the literature is sufficient. Some specific suggestions include:
- This reviewer would suggest that acronyms be minimized wherever possible to enhance readability. The terms PSU, PMPU (pg 2, ln 68) and PSMU (pg 3, ln 103) are closely related and seem to be used interchangeably.
- Discussion of the LINE social media app (pg 2, ln 58-66) seems out of place as it describes tools used for the experiments (i.e. methodology) more than literature review. Suggest moving to Materials and Methods section.
- As a point of distinction, is there a difference between phone use (i.e. text, phone calls) vs “social media”. The researchers broadly groups point-to-point interactions such as phone/texts with social media which broadly refers to larger social networks (i.e. Facebook, Twitter, etc).
Materials and Methods:
- As mentioned previously, description of the stimuli tool (LINE) should be moved to the Methods section of the paper, perhaps under “research design and procedure”.
- Should an exclusion criterial include anxious subjects. It would seem those with high levels of anxiety may bias results.
- Experimentally speaking, some explanation of the research design methodology is warranted. The design of 3 consecutive text messages with only ½ sec interval seems insufficient to allow subjects to return to baseline levels. What is the spacing between the time of the text messages and the phone call stimulus. The repetition interval (1 min) for the 3 texts + 1 call seems insufficient to allow subjects to return to baseline. Without this, it seems that the research design seems to target accumulation of anxiety over time. Further, it seems that the design of 15 texts and 5 calls within a 5-minute period is not a realistic stimulus that is representative of normal smartphone use. It would seem there is a missed opportunity to measure SCR after each set of stimuli (3 text + 1 call).
- The composition of subject enrollment seems to be ad-hoc in nature and not matched. As such, there seems to be an older cohort of males than females with p values nearing significance (0.068). This reviewer wonders if unmatched age of subjects by gender influences the results. This concern is exacerbated by the proportion of females in the group who are students (77.4%) vs males. Perhaps the correlation to SCR is influenced by age (3rd variable) which is not studied here rather than by gender.
- On pg 4, ln 161, it states the “diagnostic cut-off of 64/65” is used to assess smartphone addiction. It is unclear then why a SPAI cut-off value of 57/58 is used for assessing PSU. This lower cut-off presumably has a lower sensitivity (71%) and accuracy (70.5%). This cut-off value is not well justified and seems rather arbitrary.
Results:
- In general, t values presented is not particularly useful since this depends on the DOF and other factors. Perhaps only p-values are needed in tables.
- Table 1 is hard to read and perhaps can be better organized into different categories. The table combines many different categories (demographics, income level; smartphone use, some that may be extraneous to the research question (religious belief? Income?). Perhaps the table should be divided into 2.
- Much of the information in Table 2 (SPAI) is repeated from Table 1. Suggestion to remove from Table 1. There seem to be much more variability in the SPAI results for males vs females. This should be explained. What accounts for the significantly larger std dev?
- In Table 3, the “note” includes acronyms that are not used in the table and should be removed. It seems this note is more appropriate for Table 4. The post-test SCR stdev for females is quite high compared to the pre-test values indicating a large range in outcomes due to intervention. Some discussion of this variance warrants additional analysis. Do some females exhibit increased SCR while others do not?
- As most of the study relates to SCR measures of anxiety and PSU, there seems to be a dearth of analysis based upon SCR and these other factors (anxiety, gender, etc). It would be helpful if Table 3 (or another table) also included SCR values for anxious vs. non-anxious and low/high SPAI subjects. Aggregation of all subjects regardless of PSU/anxiety levels makes it hard to assess key effects and is not particularly useful.
- If clinically applicable SPAI thresholds are used to determine PSU, dues Table 4 change significantly? It would seem the selection of 57/58 (this study) vs. 64/65 (clinical) is rather arbitrary and not rigorous. “Note” for Table 4 copied twice.
- There seem to be significant differences in employment level of the cohort and perhaps this may affect findings (in addition to or in place of gender). Perhaps an additional factor affecting PSU and SCR is age… and maturity. These analyses are not presented and is a missed opportunity of this study. It would seem the analysis is therefore incomplete.
Discussion:
- Justification for utilization of lower SPAI scores to categorize PSU is warranted. This may influence results and findings.
It seems the discussion and conclusions are drawn from incomplete data and perhaps influenced more by the research design than the actual phenomena. Additional analysis of the data may strengthen support for the key findings.
Author Response
This manuscript studies the use of skin conductance response (SCR) for measuring anxiety with problematic smartphone use (PSU). Data is analyzed to study relationships between pre/post SCR changes and level of PSU, anxiety, and gender. PSU is a growing problem with nearly ubiquitous adoption of smartphones worldwide. There is demonstrated evidence in the literature to support the motivation for this project, namely to assess the levels of PSU/PSMU by gender and anxiety levels. SCR has been widely used to measure stress and anxiety since the 1960’s so the technology is well-developed and well-understood. The techniques deployed in these experiments is straightforward and a classical pre/post-test intervention design is implemented to study changes in SCR before and after removal of smartphones.
Q1: It would seem the consecutive nature of the stimulus is not realistic with day-to-day operation (15 texts + 5 calls in 5 minutes). The analysis conducted is very descriptive in nature and in places incomplete.
A: Our study need a biofeedback machine for SCR measurement, and every participant has his/her own job, it’s really not easy to carry out a day-to-day operation. In addition, based on the study of Lin et al., we aimed to create a withdrawal-like status during this 5-min period with counterbalance, and continuously measured participant’s SCR during this period.
Q2: Explanation is needed as to why the authors used a different SPAI cutoff level (57/58) for this study vs. clinically/diagnostically accepted levels (64/65).
A: Thank you for your recommendation. Since our purpose is to measure the SCR of and to early detect people with potential of problematic social media use, not for diagnosing those who had become smartphone addiction, we adopted 57/58 as SPAI cutoff point. The clinical workers can early refer those people with problematic social media use to an adequate unit for early intervention to prevent them from becoming addiction (Lines 111-114).
Q3: As the manuscript focuses on SCR, there seems to be limited analysis of SCR values based upon each of the key parameters (anxiety, PSU, etc). The interpretation and discussion of results may be improved with additional analysis. The key drawback of the manuscript is that the results seem to confirm what is already available in the literature without much innovation nor potential impact.
A: We have analyzed the relationship of SCR values, anxiety, and PSU (Lines 237-238 and Table 2).
Introduction: The introduction and review of the literature is sufficient. Some specific suggestions include:
1-1) This reviewer would suggest that acronyms be minimized wherever possible to enhance readability. The terms PSU, PMPU (pg 2, ln 68) and PSMU (pg 3, ln 103) are closely related and seem to be used interchangeably.
1-1) A: Thank you for your recommendation. These terms cannot be used interchangeably for their different meaning, and to make it clearer, we will use the full name of PMPU and PSMU.
The model of problematic mobile phone use (PMPU) was developed prior to the mature development and wide use of smartphones, it’s a specific term and we will keep its full name.
Problematic social media use (PSMU) here is specifically for problematic “social media” use.
1-2)Discussion of the LINE social media app (pg 2, ln 58-66) seems out of place as it describes tools used for the experiments (i.e. methodology) more than literature review. Suggest moving to Materials and Methods section.
1-2)A: Thank you for your recommendation, we have moved this paragraph to Materials and Methods section (Lines 181-184).
1-3)As a point of distinction, is there a difference between phone use (i.e. text, phone calls) vs “social media”. The researchers broadly groups point-to-point interactions such as phone/texts with social media which broadly refers to larger social networks (i.e. Facebook, Twitter, etc).
1-3)A: There are numerous social media apps, LINE is the most commonly used social media app in Asian. About 86.5% of smartphone users in Taiwan currently use the LINE app, and this proportion is much higher than other social media apps, such as Facebook, Twitter, etc (Lines 62-67 ).
Materials and Methods:
2-1)As mentioned previously, description of the stimuli tool (LINE) should be moved to the Methods section of the paper, perhaps under “research design and procedure”.
2-1)A: Thank you for your recommendation, we have moved this paragraph to the part “research design and procedure” of the section of Methods (Lines 62-67).
2-2) Should an exclusion criterial include anxious subjects. It would seem those with high levels of anxiety may bias results.
2-2)A: Anxious subjects were excluded by our exclusion criteria: pre-existing psychiatric disorders, such as anxiety disorder, depressive disorder, etc (Lines 126-127).
2-3) Experimentally speaking, some explanation of the research design methodology is warranted. The design of 3 consecutive text messages with only ½ sec interval seems insufficient to allow subjects to return to baseline levels. What is the spacing between the time of the text messages and the phone call stimulus.
The repetition interval (1 min) for the 3 texts + 1 call seems insufficient to allow subjects to return to baseline. Without this, it seems that the research design seems to target accumulation of anxiety over time. Further, it seems that the design of 15 texts and 5 calls within a 5-minute period is not a realistic stimulus that is representative of normal smartphone use. It would seem there is a missed opportunity to measure SCR after each set of stimuli (3 text + 1 call).
2-3)A: Thank you for your comments.
First, the interval between two sets of stimuli is 1/2 min, not 1/2 sec, we used the wrong word (Lines 196-197).
Second, this 5-min period was divided into five 1-min subperiod which contained stimuli for 30 sec with 3 messages and 1 call, then followed by a break for 30 sec before the next subperiod. Such design is for counterbalance of stimuli, not for returning to baseline (Lines 194-197).
Third, we have measured participants’ baseline SCR before this period, and we kept giving stimuli to every participant during this 5-min period (Lines 192-193 and Figure 1).
Participants were prohibited from using their smartphone during this 5-min period, and this design is for creating a withdrawal-like status with accumulation of anxiety (Lines 198-200).
2-4) The composition of subject enrollment seems to be ad-hoc in nature and not matched. As such, there seems to be an older cohort of males than females with p values nearing significance (0.068). This reviewer wonders if unmatched age of subjects by gender influences the results. This concern is exacerbated by the proportion of females in the group who are students (77.4%) vs males. Perhaps the correlation to SCR is influenced by age (3rd variable) which is not studied here rather than by gender.
2-4)A: Thank you for your recommendation and reminding, we divided our participants into the young (< 24 years old) and old (≥ 24 years old) groups, according Kelley’s rule of 27% and 73%, there was a significant difference on SCR (post-test SCR and mean difference SCR) between these two groups. Your recommendation is really valuable and feasible, we found that age plays an important role in our study (lines 255-257 and Table 3).
2-5)On pg 4, ln 161, it states the “diagnostic cut-off of 64/65” is used to assess smartphone addiction. It is unclear then why a SPAI cut-off value of 57/58 is used for assessing PSU. This lower cut-off presumably has a lower sensitivity (71%) and accuracy (70.5%). This cut-off value is not well justified and seems rather arbitrary.
2-5)A: Thank you for your recommendation. The SPAI cut-off value of 57/58 was used for assessing PSU in our study, and it was documented that this cut-off point is an optimal cut-off point for screening possible cases of smartphone addiction (Chen, Weng, & Su, 2003). We used the SPAI cut-off value of 57/58 was for early detecting people with the potential of problematic use.
Results:
3-1)In general, t values presented is not particularly useful since this depends on the DOF and other factors. Perhaps only p-values are needed in tables.
3-1)A: Thank you for your recommendation. We have deleted t values.
3-2)Table 1 is hard to read and perhaps can be better organized into different categories. The table combines many different categories (demographics, income level; smartphone use, some that may be extraneous to the research question (religious belief? Income?). Perhaps the table should be divided into 2.
3-2)A: Thank you for your recommendation, we have organized Table 1 and divided it into two tables. In addition, we have deleted some variables which are not closely related to our study (Table 1).
3-3)Much of the information in Table 2 (SPAI) is repeated from Table 1. Suggestion to remove from Table 1. There seem to be much more variability in the SPAI results for males vs females. This should be explained. What accounts for the significantly larger std dev?
3-3) A: Thank you for your recommendation, we have added the variables of anxiety and SPAI in the Table 3. There were some cases with extreme SPAI values which resulted in larger std dev.
3-4)In Table 3, the “note” includes acronyms that are not used in the table and should be removed. It seems this note is more appropriate for Table 4.
3-4)A: Thank you for your recommendation, we have revised the Table 3 and Table 4.
3-5)The post-test SCR stdev for females is quite high compared to the pre-test values indicating a large range in outcomes due to intervention. Some discussion of this variance warrants additional analysis. Do some females exhibit increased SCR while others do not?
3-5) A: Thank you for your reminding, we checked our data again and found that some females exhibited obviously increased post-test SCR, and some did not increase much, and this led to a large SD [Table 3 and Figure 1].
3-6) As most of the study relates to SCR measures of anxiety and PSU, there seems to be a dearth of analysis based upon SCR and these other factors (anxiety, gender, etc). It would be helpful if Table 3 (or another table) also included SCR values for anxious vs. non-anxious and low/high SPAI subjects. Aggregation of all subjects regardless of PSU/anxiety levels makes it hard to assess key effects and is not particularly useful.
3-6) A: Thank you for your recommendation, we have added two tables containing Table 2 and Table 3 and analysis among the variables which you suggested
3-7)If clinically applicable SPAI thresholds are used to determine PSU, dues Table 4 change significantly? It would seem the selection of 57/58 (this study) vs. 64/65 (clinical) is rather arbitrary and not rigorous. “Note” for Table 4 copied twice.
3-7) A: Thank you for your recommendation. The SPAI cut-off value of 64/65 was clinically used for assessing smartphone addiction. It’s an important cut-off value, but this cut-off value was not suitable for our study owing to different purpose. The SPAI cut-off value of 57/58 was used in our study to avoid missing those people with potential of problematic use.
3-8)There seem to be significant differences in employment level of the cohort and perhaps this may affect findings (in addition to or in place of gender). Perhaps an additional factor affecting PSU and SCR is age… and maturity. These analyses are not presented and is a missed opportunity of this study. It would seem the analysis is therefore incomplete.
3-8)A: Thank you for your recommendation, we have added some variables and examined the relationship among these variables. The analysis was added in the table and manuscript to make it more complete (Table 2, Table 3 and Figure 2).
4-1)Justification for utilization of lower SPAI scores to categorize PSU is warranted. This may influence results and findings.
4-1)A: Thank you for your recommendation. The SPAI cut-off value of 64/65 was clinically used for assessing smartphone addiction. It’s an important cut-off value, but this cut-off value was not suitable for our study owing to different purpose. We used cut-off vale 57/58 was for people with problematic smartphone use (PSU), not only those with smartphone.
4-2)It seems the discussion and conclusions are drawn from incomplete data and perhaps influenced more by the research design than the actual phenomena. Additional analysis of the data may strengthen support for the key findings.
4-2)A: Additional analyses were completed, and the results were shown in Table 2, Table 3 and Figure 2. We also revised our discussion and conclusion.

Round 2
Reviewer 2 Report
The revised manuscript is improved from the original version and provides additional analysis of data. However, the revision and additional data seems to contradict the major findings of the work. As alluded to previously, perhaps the data is collected in such a way (i.e. methodology) as to confound the actual scientific conclusions, perhaps even leading to incorrect conclusions by the authors. While the revision addresses some issues, serious concerns remain with the methodology, analysis, and conclusions. Specifically, these major concerns include:
1) Need to provide additional demographic details regarding study subjects including # of M/F by age group. Since the major conclusion is that age and gender affect SCR outcomes, ideally this should be a 2x2 design by age (young/old) and gender (M/F) with sufficient subjects in each group. It would seem there are insufficient subjects for this 2x2 study. Lumping all subjects in Table 1 by age or gender does not allow the audience to assess whether sufficient sample sizes are present.
2) The key results are in Table 3 and Fig 2 which seem to contradict one another. This reviewer suggests Table 3 be divided into 2x2 design to allow for analysis of M/Y, M/O, F/Y, and F/O (M=males; F=Females; Y=Young; O=Old). Currently the data by gender is lumped across age and data by age is lumped across gender. This analysis is INCORRECT. One cannot tease out the effect of these variables as constituted. Figure 2 seems to be a better depiction which suggests that there are likely no significant differences between SC_d M/Y vs. F/Y and M/O vs F/O. (note, this figure uses SCR_d whereas tables 2/3 use SC_d).
3) This reviewer does not quite understand the measure SC_d. Please clarify how SC_d is computed. Is this a difference between baseline vs. post test for each subject? If so, how can a baseline SCR of 0.34 and post-test SCR of 14.01 result in a SC_d of 0.55 as in Table 2 (non-anxiety). Should this SC_d be closer to 13..66 (give-or-take)? This metric is also used in Table 3 and Fig 2.
4) It would seem the major conclusion might be that differences is more related to age group than gender. This reviewer also notes that a 3rd variable (i.e. student/work) is not analyzed that may affect outcomes. As 37 out of 60 subjects are students and 77% of females are students, it would seem possible that the key variable is is job status and not entirely gender or age. By not analyzing this, the conclusions (age, gender) may be problematic and incorrect. Coincidentally, the research design not ensuring balanced subjects in each group (student/job) may confound the analysis.
Other corrections/suggestions include:
1) The authors try to link the stimuli with "social media" addition. As raised previously, there seems to be little social media related to this project, but rather it seems more related to smartphone use. The stimuli (ringing, buzzing) seems to have very little to do with any social media. Texting and phone calls are more linked to phone use. Just because the LINK app was used as the enabling technology does not make the study about "social media".
2) This reviewer attempted to access the Chen,Weng, Su 2003 article where the authors cited the SPAI 57/58 cutoff value for screening. It seems the 2005 paper Screening for Internet Addiction by Ko, Yen, et al is a more appropriate reference to justify this cut-off selection. This should be cited in the paper (pg 4., ln 170). Thanks for addressing this concern.
3) Table 1 is entitled "demographics of participants and gender" but include both demographics and smartphone use/behavior. Suggestion to re-title the table.
4) Suggestion in line 197 to use "30-second break" rather than ".5 min break".
5) This reviewer believes the text on lines 227-229 are reversed. 32 participants (n=32; average= > 50%) are for 30 minutes of phone use and n=30, average<50% are for duration 3-5 hours. Please re-check your results.
Author Response
1) Need to provide additional demographic details regarding study subjects including # of M/F by age group. Since the major conclusion is that age and gender affect SCR outcomes, ideally this should be a 2x2 design by age (young/old) and gender (M/F) with sufficient subjects in each group. It would seem there are insufficient subjects for this 2x2 study. Lumping all subjects in Table 1 by age or gender does not allow the audience to assess whether sufficient sample sizes are present.
1) A: Thank you for your recommendation. We have added a 2x2 design by age (young/old) and gender (M/F) in Table 1.
2) The key results are in Table 3 and Fig 2 which seem to contradict one another. This reviewer suggests Table 3 be divided into 2x2 design to allow for analysis of M/Y, M/O, F/Y, and F/O (M=males; F=Females; Y=Young; O=Old). Currently the data by gender is lumped across age and data by age is lumped across gender. This analysis is INCORRECT. One cannot tease out the effect of these variables as constituted. Figure 2 seems to be a better depiction which suggests that there are likely no significant differences between SC_d M/Y vs. F/Y and M/O vs F/O. (note, this figure uses SCR_d whereas tables 2/3 use SC_d).
2) A: Thank you for your recommendation. We have revised the Table 3 to be 2x2 design to analyze M/Y, M/O, F/Y, and F/O (M=males; F=Females; Y=Young; O=Old).
3) This reviewer does not quite understand the measure SC_d. Please clarify how SC_d is computed. Is this a difference between baseline vs. post test for each subject? If so, how can a baseline SCR of 0.34 and post-test SCR of 14.01 result in a SC_d of 0.55 as in Table 2 (non-anxiety). Should this SC_d be closer to 13.66 (give-or-take)? This metric is also used in Table 3 and Fig 2.
3) A: Thank you for your recommendation and reminding. We have revised the Table 3 and Figure 2.
4) It would seem the major conclusion might be that differences is more related to age group than gender. This reviewer also notes that a 3rd variable (i.e. student/work) is not analyzed that may affect outcomes. As 37 out of 60 subjects are students and 77% of females are students, it would seem possible that the key variable is job status and not entirely gender or age. By not analyzing this, the conclusions (age, gender) may be problematic and incorrect. Coincidentally, the research design not ensuring balanced subjects in each group (student/job) may confound the analysis.
4) A: Thank you for your recommendation. We divided participants by job into student group and employee group, and analyzed their relationship with age (young and old). We found that age was strongly positively correlated with job (p < .001) by Spearman's correlation, thus age and job cannot be analyzed together in an analysis. Future study, we need to avoid highly correlated variables while building a simple (Figure 1 and Table 2).
Other corrections/suggestions include:
1) The authors try to link the stimuli with "social media" addition. As raised previously, there seems to be little social media related to this project, but rather it seems more related to smartphone use. The stimuli (ringing, buzzing) seems to have very little to do with any social media. Texting and phone calls are more linked to phone use. Just because the LINK app was used as the enabling technology does not make the study about "social media".
1) A: Thank you for your recommendation. We have revised the term "social media" to “smartphone use”.
2) This reviewer attempted to access the Chen,Weng, Su 2003 article where the authors cited the SPAI 57/58 cutoff value for screening. It seems the 2005 paper Screening for Internet Addiction by Ko, Yen, et al is a more appropriate reference to justify this cut-off selection. This should be cited in the paper (pg 4., ln 170). Thanks for addressing this concern.
2) A: Thank you for your recommendation. We have cited the paper Screening for Internet Addiction by Ko, Yen, et al (Line 173).
3) Table 1 is entitled "demographics of participants and gender" but include both demographics and smartphone use/behavior. Suggestion to re-title the table.
3) A: Thank you for your recommendation. We have retitled the Table 1.
4) Suggestion in line 197 to use "30-second break" rather than ".5 min break".
4) A: Thank you for your recommendation. We have revised the ".5 min break" to "30-second break" (Line 198).
5) This reviewer believes the text on lines 227-229 are reversed. 32 participants (n=32; average= > 50%) are for 30 minutes of phone use and n=30, average<50% are for duration 3-5 hours. Please re-check your results.
5) A: Thank you for your recommendation. We have revised this text (Lines 229-232 and Table 1).

Round 3
Reviewer 2 Report
This reviewer would like to thank the authors for the extensive revision that addresses and clarifies many of the concerns from Round 2. A few minor corrections remain, but overall this reviewer is enthusiastic about the revision.
The re-analysis of the study data using a 2x2 dramatically enhances the rigor and quality of the research. Additionally, it improves the key findings of the work. As the 2x2 design raises concerns of small sample sizes in each cohort, the authors identify this as a limitation of the study which is appropriate.
Thank you for clarifying and correcting the SCR_d measure in Table 3 and Figure 2. This reviewer believes the SCR_d measure in Table 2 also needs to be corrected. As a correction, it seems in Table 2 and 3, the "**p<.001" should read "**p<0.01".
In Figure 2, some explanation is needed to explain the "*" and "o" notations. Are these significant differences between young-and-old by gender or between genders for corresponding age groups?
The addition of Figure 3 is a strength of the revision. Are the differences significant? Perhaps a bar chart is more appropriate for the figure. Also, adding error bars to reflect the variation is suggested.
This reviewer would like to commend the authors for addressing the various concerns and for the quality of the work and recommends acceptance with the minor revisions suggested here. Some language editing may also be needed.
Author Response
1)The re-analysis of the study data using a 2x2 dramatically enhances the rigor and quality of the research. Additionally, it improves the key findings of the work. As the 2x2 design raises concerns of small sample sizes in each cohort, the authors identify this as a limitation of the study which is appropriate.
1) A: Thank you for your suggestion that made our manuscript better and clearer.
2) Thank you for clarifying and correcting the SCR_d measure in Table 3 and Figure 2. This reviewer believes the SCR_d measure in Table 2 also needs to be corrected. As a correction, it seems in Table 2 and 3, the "**p<.001" should read "**p<0.01".
2) A: Thank you for your correcting, we have revised it (Table 2 and Table 3).
3) In Figure 2, some explanation is needed to explain the "*" and "o" notations. Are these significant differences between young-and-old by gender or between genders for corresponding age groups?
3) A: Thank you for your recommendation. We have added explanation of each notation. The significant difference was between young-and-old by gender (Figure 2).
4) The addition of Figure 3 is a strength of the revision. Are the differences significant? Perhaps a bar chart is more appropriate for the figure. Also, adding error bars to reflect the variation is suggested.
4) A: Thank you for your recommendation, we revised the Figure 3 to be a bar chart with error bars.
5) This reviewer would like to commend the authors for addressing the various concerns and for the quality of the work and recommends acceptance with the minor revisions suggested here. Some language editing may also be needed.
5) A: Thank you for your recommendation, we have revised our manuscript.
